# Effects of Recycled Fine Glass Aggregates on Alkali Silica Reaction and Thermo-Mechanical Behavior of Modified Concrete

Ibtissam Abalouch [1,2], Siham Sakami [3], Fatima-Ezzahra Elabbassi [3] and Lahcen Boukhattem [1,2,*]

1   LMPEQ, National School of Applied Sciences, Safi, Cadi Ayyad University, Marrakech 40000, Morocco; ibtissam.abalouch@ced.uca.ma
2   EnR2E Laboratory, National Center for Studies and Research on Water and Energy, CNEREE, Cadi Ayyad University, Marrakech 40000, Morocco
3   L3G, Faculty of Science and Techniques, Cadi Ayyad University, Marrakech 40000, Morocco; s.sakami@uca.ma (S.S.); fz.elabbassi@uca.ma (F.-E.E.)
*   Correspondence: l.boukhattem@uca.ma

**Abstract:** The objective of this work is to develop a new composite material by substituting sand with recycled waste glass (RWG). Different volume percentages of RWG varying from 0 to 50% were incorporated into concrete, with maximum size that did not exceed 1.25 mm. The microscopic characterization by scanning electron microscopy SEM-EDS and optical microscopic test, as well as the durability against alkali silica reaction (ASR) test, were performed respectively to visualize the morphology and assess the damage caused by ASR. Furthermore, the mechanical and thermophysical properties measurements were carried out. The results of microscopic characterization showed the presence of cracks inside a minority of glass particles due to ASR, and ASR test indicated that expansion activity remained well below the limit expansion value of 0.15%. The obtained results also showed that, at 28 and 90 days of curing, compressive strength increased respectively by up to 1.63% and 29% for 20% of the incorporated RWG volume rate in concrete; however, beyond this rate it diminished receptively by 30% and 3.2%. This improvement with curing age was attributed to pozzolanic reaction. Regarding density, it reduced by around 5%. Furthermore, thermal conductivity and thermal effusivity decreased respectively by 36% and 8.06% at dry state and they dropped respectively by 44% and 21.28% at saturated state, related to reference concrete RC. It is therefore feasible to substitute high amount of natural sand with RWG to obtain new composite that may be successfully used as structural material in construction building.

**Keywords:** recycled waste glass; concrete; durability; compressive strength; thermal conductivity

## 1. Introduction

In 150 countries around the world, building sector consumes annually 28.7–32.8 billion tons of virgin aggregates such as rock, stone, gravel, and sand [1]. Due to increased demand of sand, sand quarries have become scarce, and nowadays, many construction contractors are illegally stripping beaches and dunes to satisfy their needs. In Morocco, half of the sand used each year in construction sector, about 10 million cubic meters, is illegally extracted [2]. This increasing amount of extracted sand has significant environmental and social impacts. For these concerns, the United Nation of Environment Program UNEP indicated some already existing solutions such as avoiding consumption through reducing over-building and over-design, reducing impacts through implementing existing standards and best practices, and using recycled and alternative materials to sand in the construction sector. It also proposed different actions, of available solutions and subsequent consultations, to be quickly implemented to tackle sand extraction impacts [1]. At national scale, Morocco, with a coastal board of 3446 km, has taken a big step to protect it by adopting a legal process. However, these measures remain insufficient [3].

On the other hand, the world population growth and the continuous lifestyle improvement by using inorganic materials such as plastics, paper, aluminum, and glass instead of organic ones [4] lead to an increased amount generation of solid waste. For instance, a world glass industry produced a volume of 140 million tons in 2016 compared to 89.4 million tons in 2007 [5,6]; a huge amount of this produced glass was turned into solid waste. Faced with this situation, many researchers have valorized the waste glass to be reused in building sector by integrating it into concrete and mortar. Furthermore, Refs. [7,8] reported that mechanical performance of eco-composite material depends on size, quantity, and colors of recycled glass; Refs. [9,10] indicated that the fineness of the glass aggregates improved the mechanical resistance of the composite. Refs. [11–13] showed that the reinforcement of concrete with 20% of RGW had a positive effect on compressive strength as it was enhanced by up to 20%. To produce a sustainable eco-concrete, M. Olofinnade et al. [14] recommended that the optimum glass content in concrete should be of 25%. Adaway and Wang [15] and Malik et al. [16] found that the optimal percentage of glass waste giving the maximum values of resistance to compression and bending was 30%. Furthermore, Tamanna et al. [17] reported that after 56 days of curing, no adverse reduction in strength was noticed for 60% of sand replacement with waste glass. Concerning the effect of waste glass color on concrete's mechanical properties, park et al. [18] found that it had no influence on these quantities. However, De Castro and De Brito [19] found that at 28 days of curing, the reuse of waste glass as fine (<4 mm) or coarse aggregates (<11.2 mm) decreased the concrete strength. Similarly, Polly et al. [20] showed that compressive strength was reduced by 50% when 15% of sand in concrete was replaced by glass aggregate coarser with size less than 1.5 mm; this is due to the extremely poor shape, poor surface characteristics, and high friability of the used glass. Tan and Du [21] mentioned that the use of clear glass as sand in mortar showed the poorest mechanical performance. In addition to mechanical properties determination, some authors have dealt with thermal characterization measurement. In recent years, Alani et al. [22] have reported that the thermal conductivity of screed with 100% of glass aggregate was almost 50% lower than that of sand screed. In 2013, the thermal conductivity with 50% of RGW with size less than 4 mm decreased by around 18% and 22% respectively at dry and wet states compared to reference concrete RC [23]. Guo et al. [24] showed that when sand was entirely replaced by RWG, the reduction in thermal conductivity was more significant for architectural mortars. Additionally, ref. [25] were interested in the use effect of the recycled glass on the lightweight concrete, and they showed that the integration of 45% of glass as sand in concrete led to the thermal conductivity reduction by about 12% and 16% with respect to RC respectively at ambient and elevated temperature 600 °C. Chung et al. [26] showed that the replacement of total sand with crushed glass (0–4 mm) in lightweight concrete specimen decreased its thermal conductivity by around 60%. In 2019, Shuqing Yang et al. [27] reported that regardless of the particle size, the thermal conductivity and density of the dry-mix blocks made with glass aggregates were significantly lower than the ones made with river sand.

Moreover, concrete modified with RWG is characterized by creation of two different reactions, with opposite effects. First, the substitution of sand with glass grains in the concrete gives rise to beneficial pozzolanic reaction between the ground glass and cement past, result in formation of the compounds with binding properties. These compounds are amorphous gels of the C-S-H type, with Ca/Si ratios generally much lower than those of the C-S-H in cement [28]. Shi et al. [29] found that finely ground glass powders exhibited very high pozzolanic activity. Idir et al. [30] showed that after 90 days of curing for 20% of glass, the pozzolanic activity became substantial when the transition fineness was around 30 $m^2.kg^{-1}$ (140 μm). Shayan and Xu [31] observed that when 30% of glass powder was replaced by sand, the strength was significantly greater than that of the control mixture due to beneficial pozzolanic activity. Second, the reinforcement of concrete with RWG generated a detrimental alkali silica reaction (ASR), known as concrete cancer, by forming amorphous gels between the reactive silica in glass and alkali ions in cement paste. The

formed alkali silica provoked cracks in the elaborated composite [32]. The effectiveness of this phenomenon grew with the size increase of the glass particles [7,9,33]. Furthermore, the ASR in concrete with glass could also be caused by the micro residual cracks inside recycled glass particles, generated during the glass bottle crushing operations [34–36]. To attenuate the deleterious alkali silica reaction, some authors proceeded to add suppressors into concrete: [37] added fly ash, [38] dealt with silica fume, and [39] were interested in rice husk ash.

To the best of our knowledge, this is the first scientific research which simultaneously investigated the durability against alkali silica reaction and mechanical and thermal properties of concrete reinforced with different volume percentages of RWG varying from 0 to 50%. Two major issues were addressed, namely the scarcity of sand and the great potential of waste glass that is misused. Durability against alkali-silica reaction was assessed by autoclaving method, SEM-EDS, and optical microscopic to examine the concrete-based composite stability. Compressive and flexural strengths were determined at 28 and 90 days of curing. Furthermore, measurement series of thermal conductivity and thermal effusivity were carried out at dry and saturated states.

## 2. Test Samples and Experimental Methods Description

### 2.1. Materials

The used sand and gravel (coarse aggregates) in this work were extracted from Ghartour quarry in Oued Tensift, 40 km away from Marrakech city. Their maximum sizes are respectively of 5 mm and 20 mm. Recycled waste glass (RWG) was obtained from different glass wastes with different colors; its mixture was composed of 40% of green color, 30% of topaz, 25% of white glass, and 5% of blue glass as shown in Figure 1. To reduce the ASR reaction, the diameter of glass waste particles was chosen less than 1.25 mm [7]. The granulometric analysis carried out according to NF P18-560 [40] and chemical composition of these materials are respectively given in Figure 2 and Table 1. The used binder is an ordinary Portland cement CPJ 45 which is equivalent to CEM I 42.5. It has a Blaine fineness of 3941 $cm^2.g^{-1}$ and a compressive strength of 33.2 Mpa at 28 days. Finally, a superplasticizer of Master Glenium 118 M based on polycarboxylate ether with density of $1.090 \pm 0.005$ $g.cm^{-3}$ and mass content of 0.8 to 2% relative to cement was used to obtain concrete with high quality.

Table 2 gives different parameters values such as flakiness coefficient (FC), superficial cleanness (SC), and Los Angeles abrasion test (LA) determined for coarse aggregate; as well as fineness modulus of sand, and equivalent of sand (ES) determined to evaluate its property and compliance. Specific weight, water absorption, and bulk density of all aggregates were also determined.

### 2.2. Specimens Preparation

The test samples were prepared by substituting sand with different RWG volume percentages varying from 0% to 50%. Reference concrete (RC) was prepared first by mixing cement, sand, gravel, water, and superplasticiser. Next, the concrete-based composite was obtained by replacing the sand with RWG. Five composites, labeled as RC, CRWG10, CRWG20, CRWG30, and CRWG50, were manufactured. The water-cement and superplasticizer-cement mass ratios were modified in this work for each mixture operation so as to obtain a slump value of 50 mm. The mass of each constituent in 1 $m^3$ of each studied composite is given in Table 3.

Samples preparation consists in mixing all the dry constituents of concrete for 5 min in cement mixer (Figure 3a). Subsequently, water and adjuvant were progressively added until a desired slump value by Abrams cone according to NF P 18-451 was obtaining [41] (Figure 3b,c). Then, the mixture was immediately poured in molds (Figure 3d).

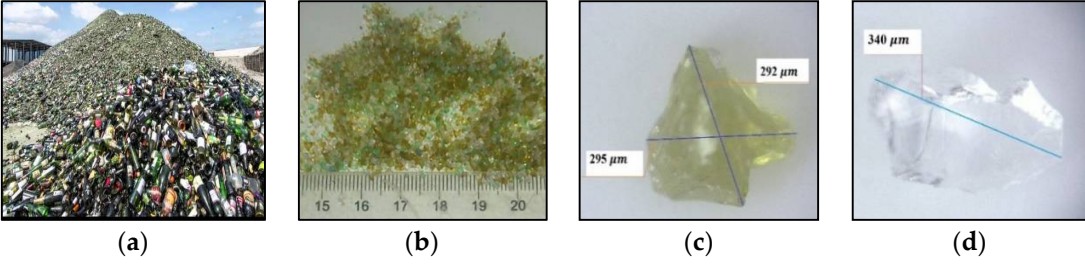

**Figure 1.** Waste recycled glass: (**a**) waste glass, (**b**) RWG particles with different colors, (**c**,**d**) size of the RWG particle.

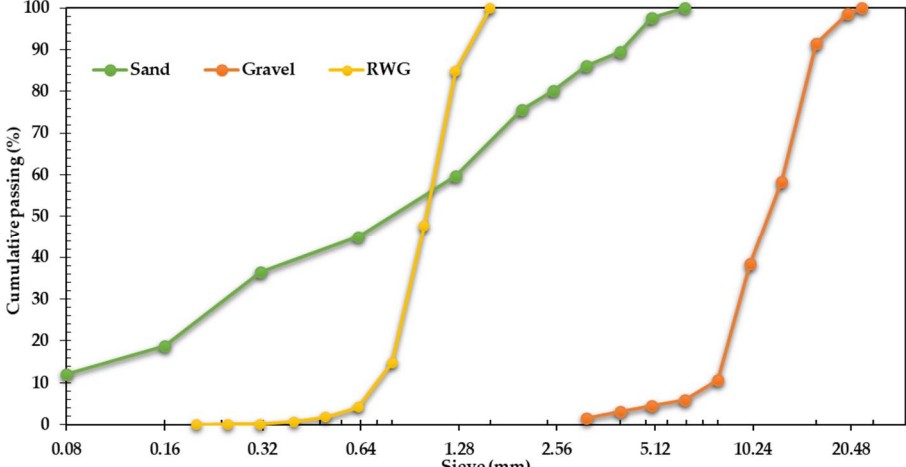

**Figure 2.** Granulometric analysis of sand, gravel, and recycled waste glass.

**Table 1.** Chemical components of sand and RWG determined by XRF elemental analysis.

| % | $SiO_2$ | $Al_2O_3$ | $Fe_2O_3$ | CaO | MgO | $K_2O$ | MnO | $TiO_2$ | $P_2O_5$ | $Na_2O$ | $Cr_2O$ |
|---|---|---|---|---|---|---|---|---|---|---|---|
| **RWG** | 50–70 | 5–25 | <1 | 5–25 | 1–5 | 1–5 | <0.5 | <0.5 | <0.5 | 5–25 | <0.5 |
| **Sand** | 68.51 | 13.55 | 2.83 | 1.62 | 2.64 | 3.95 | - | 0.7 | 0.16 | 2.72 | - |

**Table 2.** Physical properties of different used aggregates.

| Nature | Gravel | Sand | RWG |
|---|---|---|---|
| Flakiness coefficient FC | 13 | - | - |
| Superficial cleanness SC (%) | 0.7 | - | - |
| Los Angeles abrasion test LA (%) | 21 | - | - |
| Fineness modulus | - | 2.62 | - |
| Equivalent of sand ES | - | 75 | - |
| Specific weight ($T.m^{-3}$) | 2.63 | 2.68 | 2.5 |
| Water absorption (%) | 0.5 | 2.5 | 0 |
| Bulk density ($kg.m^{-3}$) | 1455 | 1500 | 1300 |

**Table 3.** Mass of each constituent in 1 $m^3$ of each formulation of concrete reinforced with RWG.

| Composite | Materials (kg) | | | | Water (kg) | Superplasticizer (kg) |
|---|---|---|---|---|---|---|
| | RWG | Gravel | Sand | Cement | | |
| **RC** | - | 995.56 | 797.10 | 400 | 180 | 6.80 |
| **CRWG10** | 74.35 | 995.56 | 717.39 | 400 | 172 | 6.08 |
| **CRWG20** | 148.71 | 995.56 | 637.68 | 400 | 162.79 | 5.04 |
| **CRWG30** | 223.06 | 995.56 | 575.97 | 400 | 161.66 | 5.96 |
| **CRWG50** | 371.78 | 995.56 | 398.55 | 400 | 172.72 | 6.4 |

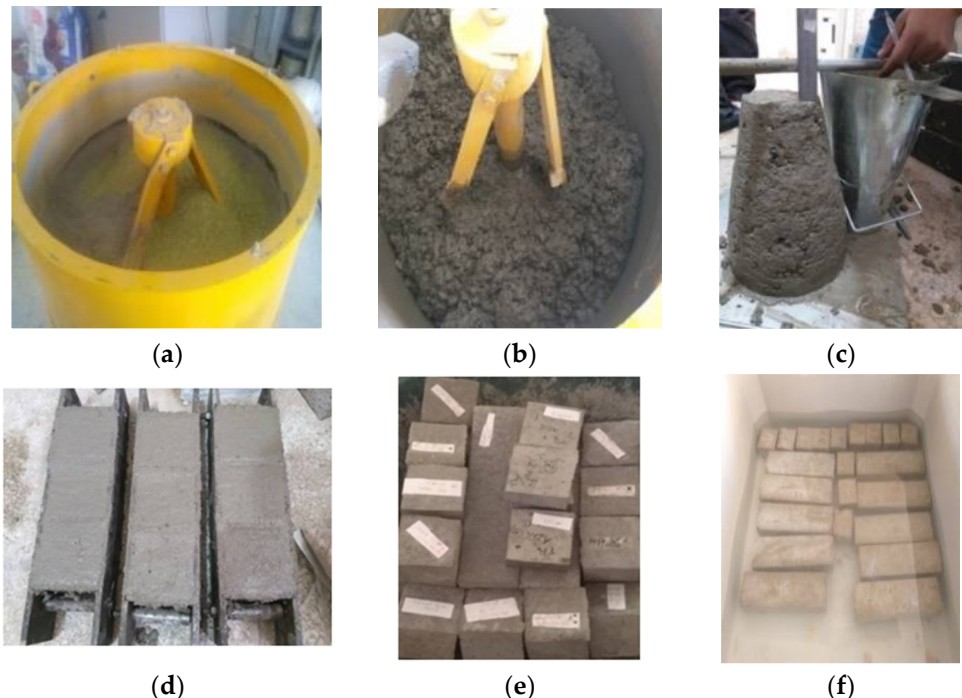

**Figure 3.** Preparation of specimen: (**a**) cement mixer machine, (**b**) mixing of concrete, (**c**) slump measurement, (**d**) filled molds, (**e**) samples after demolding operation, and (**f**) specimens in fresh water.

The compression and flexion samples are respectively cubic with dimensions of $10 \times 10 \times 10$ cm$^3$ and prismatic with dimensions of $10 \times 10 \times 40$ cm$^3$ (Figure 3e). All specimens were kept in the mold for 24 h at $20 \pm 2$ °C, according to NFP18-404 [42]. Afterwards, they cured for a period of 27 days in fresh water at 20 °C. Some of these samples were kept in the environment for more 62 days, as shown in Figure 3f.

### 2.3. Experimental Methods Description

2.3.1. Microscopic Characterization

Microscopic characterization by TESCAN ORSAY (VEGA3) Scanning Electron Microscopy (SEM) with Energy Dispersive X-ray Spectroscopy (EDS) and optical microscope EUROMEX ISCOPE were manipulated to visualize the morphology of samples. The analysis of different spots in the glass surrounding the concrete-based composite was carried out by SEM-EDS. It is a procedure in which characteristic X-rays generated from the electron beam-sample interaction are analyzed to provide elemental composition of the sample. Their results are in the form of spectra (histograms) in which individual elements can be identified. It provides quantitative chemical characterization of the samples.

The observation of adhesion between RWG and cement paste and the evaluation of damage caused by ASR reaction was achieved by EUROMEX ISCOPE. This technique is supplied with a pair of $10\times/20$ mm (Ø 30 mm tube) eyepieces. It has also an intensity adjustable 3 W illumination with internal 100–240 V power supply. For this purpose, thin blade samples with thickness of 30 μm for MRWG50 (Mortar reinforced with 50% of RWG) and CRWG50 were prepared, as shown in Figure 4. The preparation of these thin blades consisted, in a first step, in sawing a piece of a control sample with diamond saw until reaching a thickness of 2 to 1 mm. Thereafter, a blade glass was fixed on the sample's flat surface by a lapping machine and pressed at 80 °C during 1 h. The obtained blade was then maintained in the machine by an aspiration system to be cut in a parallel plan to glass blade at thickness of 2 to 1 of tenth of millimeter. Finally, the lapping device lasted one hour to rotate and abrade the blade with abrasive powder of silicon carbide until achieving a thickness of 30 μm.

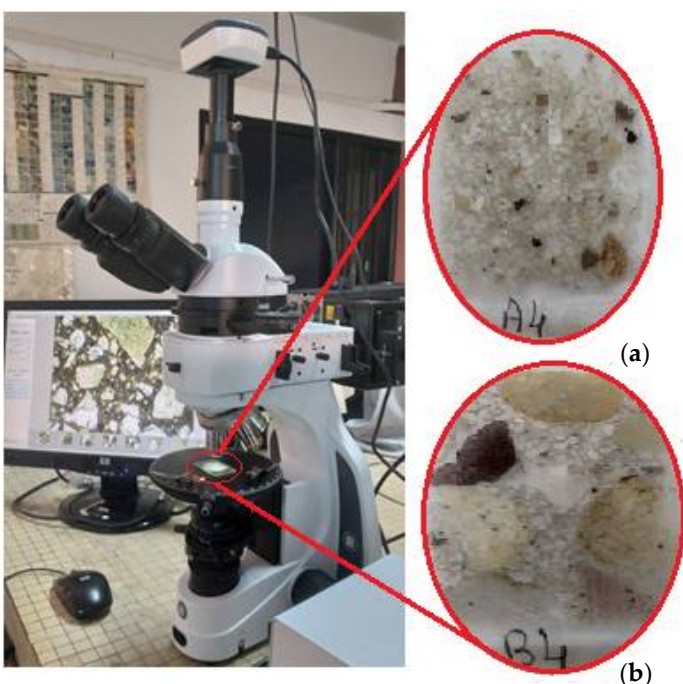

**Figure 4.** Euromex iscope optical microscopic of thin blade samples: (**a**) after ASR test of MRWG50 and (**b**) at 90 days of cured CRWG50.

### 2.3.2. Durability against Alkali-Silica Reactivity

Alkali-silica reactivity (ASR) autoclaving test qualifies with certainty the no reactivity of different aggregates and it showed the best performance [38,43]. This test was carried out in this work in accordance with NF P-P18-594 [44]. First, sand with granularity between 0.16 mm and 5 mm was obtained. Second, this sand was washed and dried at $(80 \pm 5)$ °C; and its particles were classified according to their sizes, as indicated in Table 4. Afterwards, NaOH was added to water till 4% mass content of $Na_2O$ related to mass of cement. Each composite was then prepared by mixing different components as given in Table 4 (Figure 5a). The obtained mixture was immediately poured in three prismatic molds where on both their sides were fitted stainless steel studs to measure dimensional variations (Figure 5b,c). Furthermore, these molds, each with dimensions of $40 \times 40 \times 160$ mm$^3$, were placed in a chamber with relative humidity of 90% and temperature of $(20 \pm 1)$ °C for 24 h. After demolding operation (Figure 5d), the mortar specimens were immersed in clean tape water at $(20 \pm 1)$ °C for 48 h. After this time, the initial lengths $L_{0i}$ ($i = 1$, 2, 3) were measured with comparator with an accuracy of 0.001 mm (Figure 5e,f). Next, the test samples were put in autoclave (Figure 5g) with a relative pressure of $(0.15 \pm 0.01)$ MPa and temperature of $(127 \pm 2)$ °C during 5 h $\pm$ 30 min. Finally, after 18 h $\pm$ 30 min of rest in ambient temperature (Figure 5h), the final lengths of these samples were measured $L_{1i}$ ($i = 1$, 2, 3).

**Table 4.** Formulation of each mortar-based composite for ASR test.

|  |  |  | RM | MRWG10 | MRWG20 | MRWG30 | MRWG50 |
|---|---|---|---|---|---|---|---|
| Composite Constituents | Sand classified with grain size (g) | 0.16–0.31 | 120 | 108 | 96 | 84 | 60 |
|  |  | 0.31–0.63 | 120 | 108 | 96 | 84 | 60 |
|  |  | 0.63–1.25 | 300 | 270 | 240 | 210 | 150 |
|  |  | 1.25–2.50 | 300 | 270 | 240 | 210 | 150 |
|  |  | 2.50–5.00 | 360 | 324 | 288 | 252 | 180 |
|  | Waste glass (g) |  | 0 | 111.94 | 223.88 | 335.82 | 559.70 |
|  | Cement (g) |  | 600 | 600 | 600 | 600 | 600 |
|  | Water with NaOH Addition (L) |  | 300 | 300 | 300 | 300 | 300 |

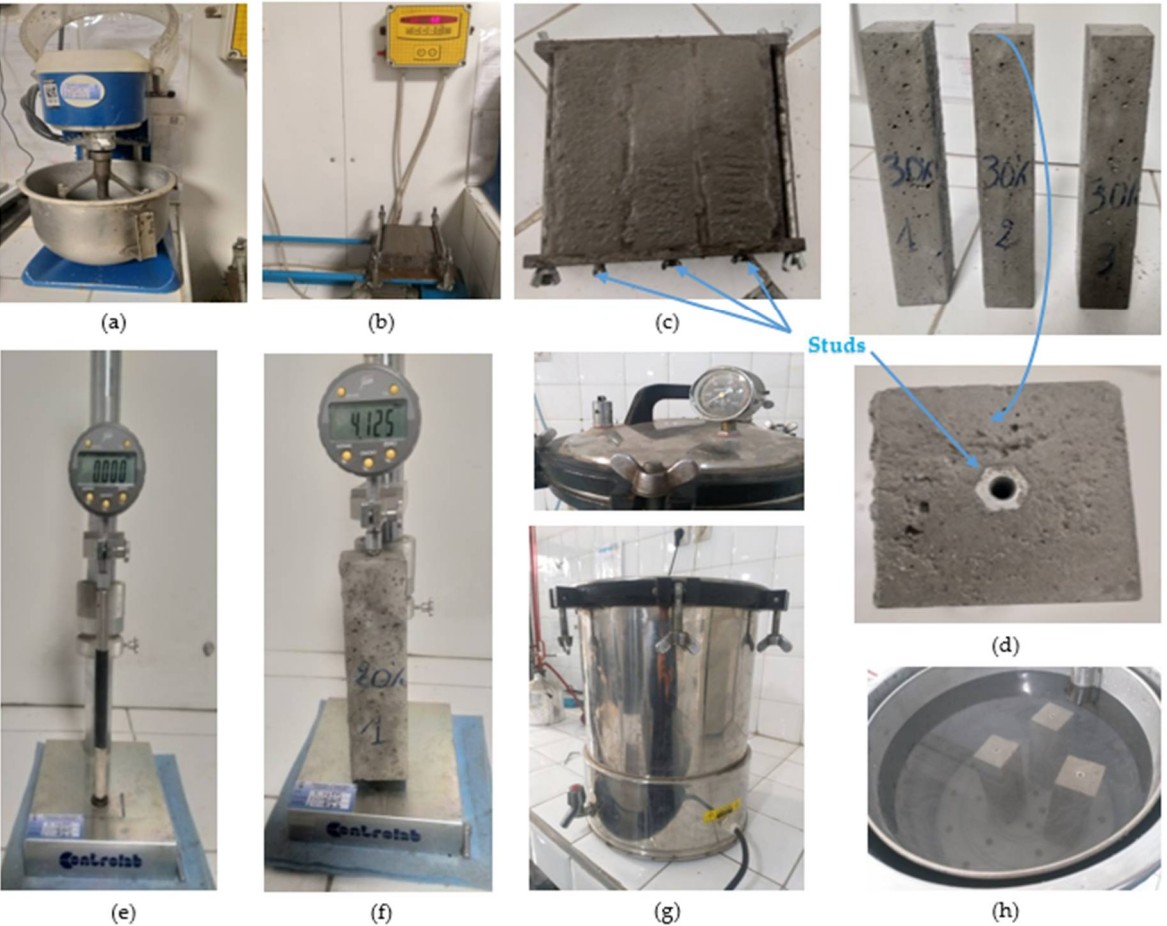

**Figure 5.** Experimental details of durability test against ASR: (**a**) mixer, (**b**) vibrating table, (**c**) prismatic molds, (**d**) samples after demolding operation, (**e**) numeric comparator with calibration bar, (**f**) measure of elaborated sample, (**g**) autoclave, and (**h**) samples after autoclave opening.

The relative strain (%) was obtained as being the arithmetic mean value of different strains.

$$\overline{\varepsilon} = \sum_{i=1}^{3} \varepsilon_i \tag{1}$$

where:

$$\varepsilon_i = \frac{L_{1i} - L_{0i}}{L_0}, \ L_0 = 160 \ \text{mm} \tag{2}$$

### 2.3.3. Mechanical Properties

Compressive strength of manufactured samples was determined at the age of 28 and 90 days of cure using the servo plus evolution E1177 device; its speed is of 0.5 MPa.s$^{-1}$ and departure force of 10 kN (Figure 6a).

The compressive strength of cubic shape sample with $a = 10$ cm requires an adjustment according to NF P 18-325-1 [45]:

$$F_{c,\ cub} = 0.97 \times R \ \text{If R} > 50 \ \text{MPa} \tag{3}$$

$$F_{c,\ cub} = R - 1.5 \ \text{If R} < 50 \ \text{MPa} \tag{4}$$

R is the obtained compressive strength value read on the device.

The flexural strength was measured according to NFP 18-433 [46] by using Controls device with a speed of 0.04 MPa.s$^{-1}$ at 28th day of specimen's cure. Its measurement method is displayed in Figure 6b,c.

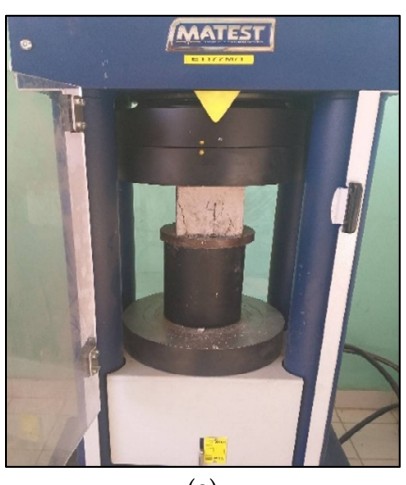
(a)

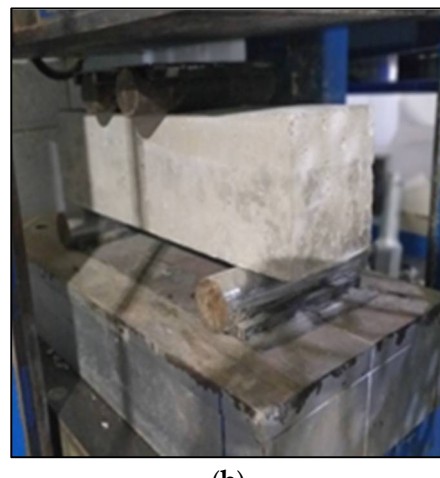
(b)

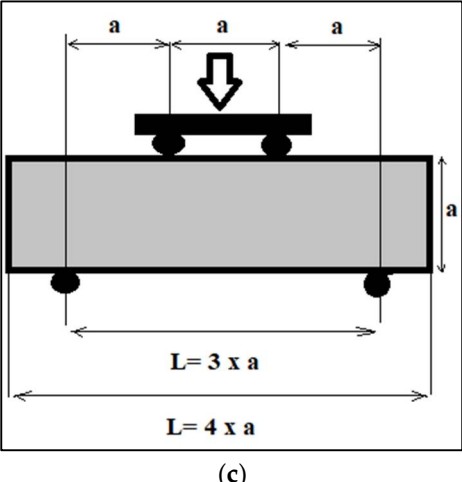
(c)

**Figure 6.** Mechanical tests: (**a**) compressive strength, (**b**,**c**) flexural strength.

The flexural strength $F_f$ corresponds to the maximum load recorded during the test and it can be calculated as follows:

$$F_f = \frac{30F}{a^2} \tag{5}$$

where $F$ is the breaking load.

2.3.4. Thermo-Physical Properties

To determine the thermo-physical properties of the concerning samples, a hot wire conductivimeter device (FP2C) was manipulated to measure thermal conductivity ($\lambda$) and thermal effusivity ($E_f$) of a material using respectively the hot wire method and hot plane method as shown in Figure 7. The reliability of the FP2C device was ensured by comparing the thermal conductivity results obtained by this method to the ones obtained by boxes method for two different samples [47]. Density of composite was also determined according to NF P 435 [48]. These quantities were measured six times for each sample so as to determine their uncertainty. Once these three quantities are measured, the thermal diffusivity ($\alpha$) and volumetric heat capacity $\left(\rho C_P\right)$ of the sample can be calculated as follows:

$$\alpha = \left(\frac{\lambda}{E_f}\right)^2 \tag{6}$$

$$\rho C_P = \frac{E_f{}^2}{\lambda} \tag{7}$$

The measurements were carried out first on samples at saturated state (directly after 28 days of curing), then at dry state after having been dried for 72 h at 105 °C.

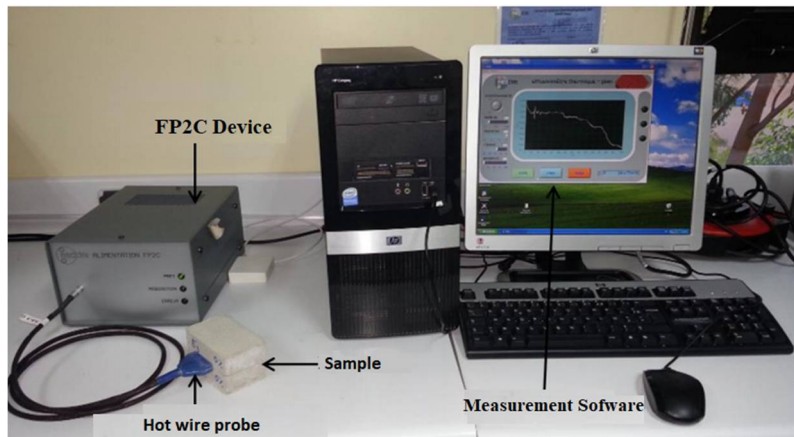

**Figure 7.** FP2C device used to measure thermal conductivity and thermal effusivity of the studied samples.

## 3. Results and Discussion

### 3.1. Microscopic Characterization Analysis

Figure 8 shows SEM-EDS micrographs of modified mortar (MRWG50) with 50% of sand replacement with RWG. The region indicated in Figure 8a was deeply analyzed by identifying four spots as it was visualized at length scale 100 μm (Figure 8b), and the found results are depicted in Table 5. The identified Spot 1 indicates chemical components of glass similar to those given in Table 2, Spot 2 shows an important content of calcium to Silicon (Ca/Si) of 5.4 which represents chemical elemental compositions of cement paste products, Spots 3 and 4 in the surrounding glass are characterized by Ca/Si values of 1.51 and 1.87 respectively, representing the chemical constituents of calcium-silica-hydrate C-S-H different from the ones of ASR gel. In this regard, many studies [35,49,50] found that ASR gel is characterized by an amount of calcium Ca less than that of silicon Si; which does not match our results found around the glass. This latest finding indicates that the RWG incorporation in the studied samples didn't produce deleterious alkali-silica reaction.

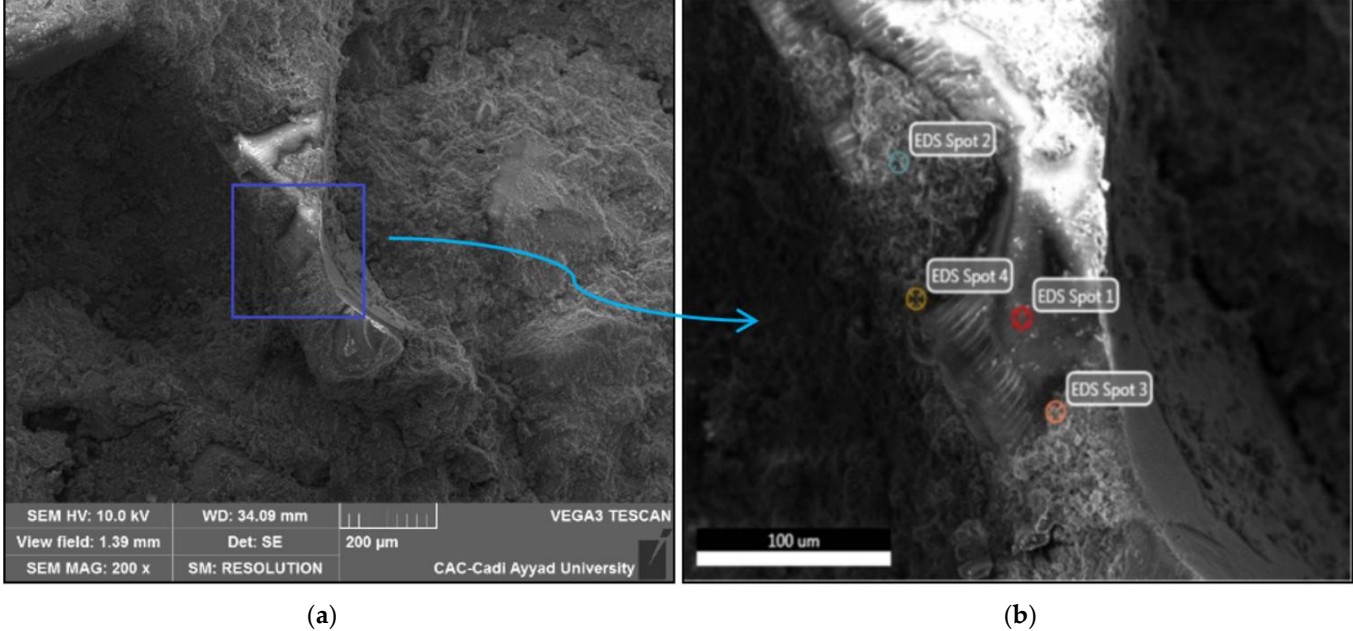

**Figure 8.** SEM-EDS micrographs of MRWG50: (**a**) SEM picture at length scale 200 μm and (**b**) Spots of EDS imaging.

**Table 5.** SEM-EDS analysis of mortar incorporated with 50% volume content of RWG (MRWG50).

| Spots/ASR gel | Chemical Components (Weight %) | | | Ca/Si (W %) Present Work | Ca/Si (W %) [51] | CaO/SiO$_2$ (W %) [50] | CaO/SiO$_2$ (W %) [35] |
|---|---|---|---|---|---|---|---|
| | Ca | Si | Na | | | | |
| spot 1-RWG | 6.85 | 29.91 | 12.15 | 0.21 | | | |
| spot 2-Cement paste | 81 | 15 | 0 | 5.4 | | | |
| spot 3-Glass-paste interface | 18.99 | 12.55 | 4.03 | 1.51 | | | |
| spot 4-Glass-paste interface | 44.74 | 23.84 | 2.39 | 1.87 | | | |
| ASR gel | | | | | 0.58 | 0.42 | 0.27 |

To further visualize the damage caused by ASR, a microscopic characterization by optical microscope technique was carried out. This technique is among the accurate one to know the degradation state associated to ASR [52]. Figure 9 presents optical microscope characterization of MRGW50. It can be observed that a minority of glass particles were cracked while the majority was not. These cracks were caused by ASR due to penetration of ions inside micro-cracks generated already during the glass bottle crushing operations [35,50,51].

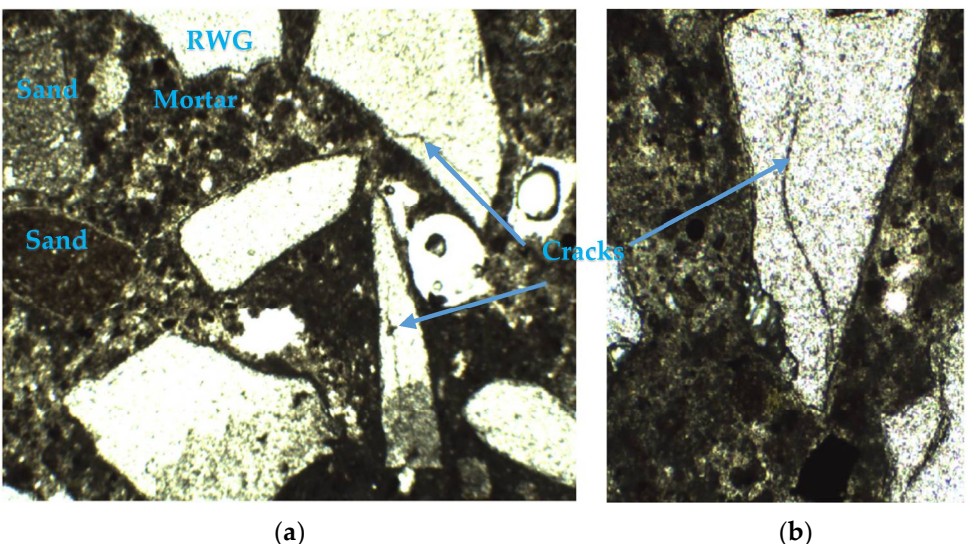

(**a**)  (**b**)

**Figure 9.** Optical microscopic view of MRGW50 after autoclaving test of durability against ASR: (**a**) at 50× magnification; (**b**) at 100× magnification.

This result matches the one reported by Rajabipour et al. [35], and Guo et al. [51] who both showed that ASR gel occurs in the micro-cracks already existing in the glass particles. It means that the only particles with micro-cracks are attacked by ASR.

The microscopic view of the bond between glass particles RWG and cement paste in concrete sample CRWG50 after 90 days of curing was also carried out. Figure 10 shows that strong bonds were created between RWG and cement paste and no voids were observed between the two elements. These bounds were created as a result of the pozzolanic CSH due to reaction of dissolved silica with the available portlandite [28,35].

*3.2. Durability against Alkali-Silica Reactivity*

Figure 11 shows the expansion of mortar-based composite containing different ratios of RWG with different colors. At first glance, it can be clearly observed that the expansions relating to the ASR increased with the increasing percentage of RWG in mortar. This behavior was reported by Topcu et al. [53] and Yamada et al. [54] who studied the effect of color, size, and mass content of recycled waste glass incorporated in concrete. They showed respectively that green glass aggregate presents the best performance for 25% glass content

regarding ASR resistance, and the expansion falls below 0.1% when the maximum dosage of glass particles with maximum size of 2.36 mm is 10%.

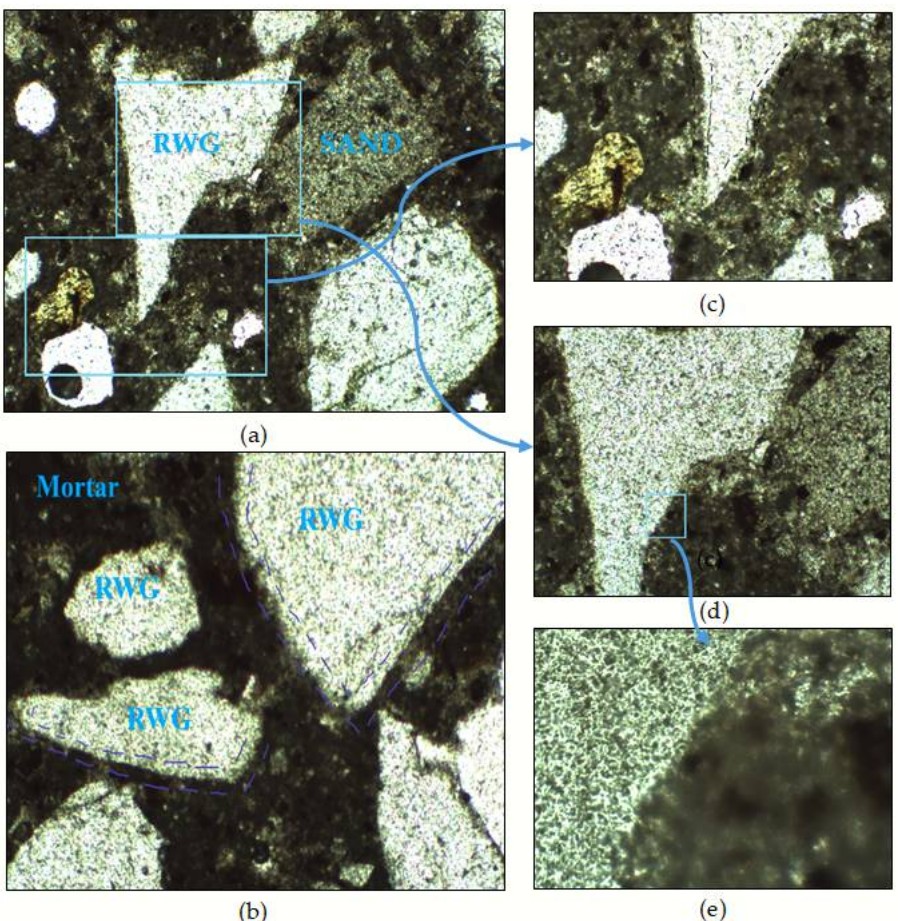

**Figure 10.** Optical macroscopic view of CRWG50 after 90 days of curing: (**a**,**b**) at magnification 100×; (**c**,**d**) at 200× magnification; (**e**) at 500× magnification.

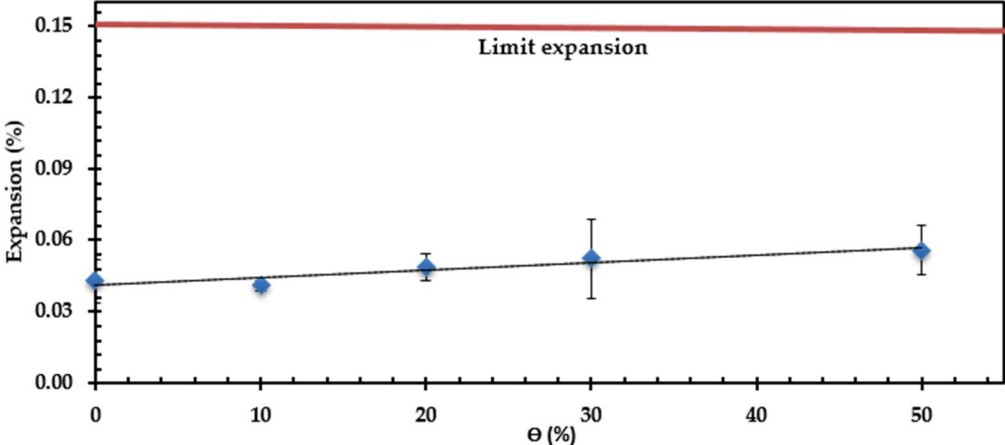

**Figure 11.** Expansion of different mortar-based composites reinforced with different volume contents of RWG, compared to limit expansion value of 0.15%.

Therefore, the expansion of mortar reinforced with different volume contents of RWG is considerably lower than the maximum value of 0.15% prescribed in the standard specifications [44]. Indeed, the expansions of RM, MRWG10, MRWG20, MRWG30, and

MRWG50 are lower to limit expansion value respectively by 71.46%, 72.76%, 67.50%, 65.14%, and 62.87%. The result of MRWG50 indicates that ASR increased with the RGW amount increase in concrete, but it remained below the limit expansion value. This finding is in line with the small amount of cracked particles noticed in Figure 9.

The relationship between the expansion and the volume percentage of RWG, for the studied composite materials, can be expressed as a linear line, with a correlation coefficient of $R^2 = 0.88$.

$$\varepsilon = 0.0003\theta + 0.0414 \tag{8}$$

### 3.3. Experimental Results of Mechanical Properties

#### 3.3.1. Compressive Strength

The compressive strength variations with RWG at 28 and 90 days of curing are presented in Figure 12. At 28 days the compressive strength increased slightly by 1.49% and 1.63% with the incorporation of 10% and 20% of RWG respectively, compared to RC. This slight increase may be attributed to the geometry and size of the RWG compared to the rounded natural sand which improves adhesion between cement paste and glass particles [14,15]. However, it decreased by 12.43% and 29.45% respectively for CRWG30 and CRWG50. This drop is due to the RGW amount increase in concrete which leads to a low fineness modulus [18], and to the slow pozzolanic reaction that occurs at a later stage and leads to low adherence between cement paste and glass particles for the highest volume content of RWG in the mixture [29].

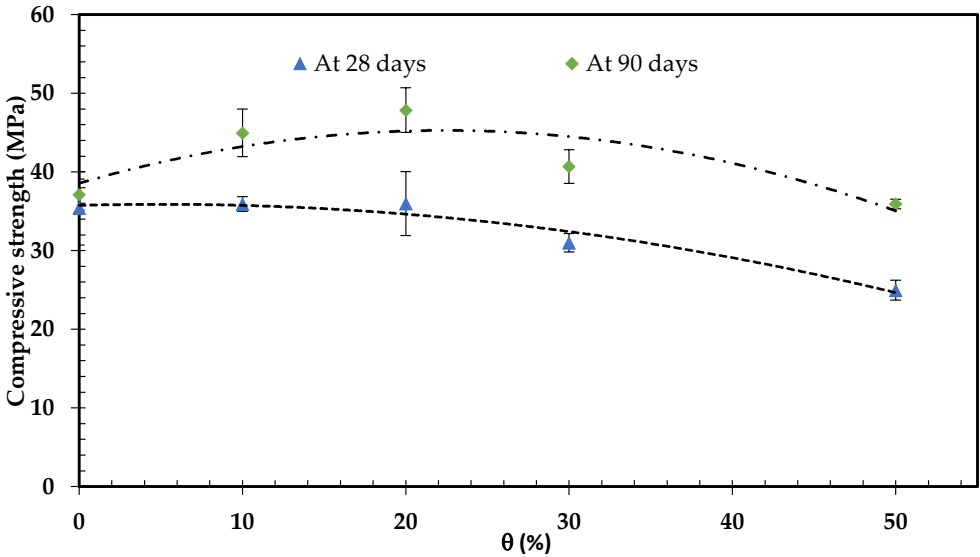

**Figure 12.** Compressive strength influence of concrete reinforced with different volume percentages of RWG.

At 90 days of curing, the compressive strength increased by 21.11%, 28.89%, and 9.58% respectively with addition of 10%, 20%, and 30% of RWG in concrete; the maximum value was observed for RWG20, whereas it decreased slightly for RWG50 by 3.22% compared to RC. As an overall observation, compressive strength evaluated better with cure age, mainly the one of CRWG50 which reached 30.51% between 28 and 90 days of curing compared to 4.68% of RC. It means that at 28 days of curing, RC achieved 95.32% of its final strength; however, the compressive strength of RWG50 reached only 77% of its strength at 90 days. This finding is due to the pouzolanic effect of glass in concrete, leading to slow growth in compressive strength [55].

From the microscopic view at 90 days of curing (Figure 10), it can be noticed that the RWG particles are well bound to the ones of cement. These bounds were produced by C-S-H from the slow pozzolanic reaction between silica in glass and alkali ions in cement [28,35].

### 3.3.2. Flexural Strength

Figure 13 shows the flexural strength variation with volume content of RWG in concrete at 28th day of curing. With 50% volume content of RWG in concrete, it can be observed that the flexural strength decreased by about 34.10% with respect to RC. The variation tendency is similar to the one observed for compressive strength at 28th day of curing, namely between 0 to 20% (Figure 12). From 20 to 50% of RWG, one can notice that the flexural strength decreased significantly due to the adherence between cement paste and glass particles that becomes lower as the RWG is more and more incorporated into concrete. The same result was reported by several researchers [29,30,32]; for instance, Park et al. [18] found that, with addition 30% of volume percentage of RWG, the flexural strength decreased by 11.3%.

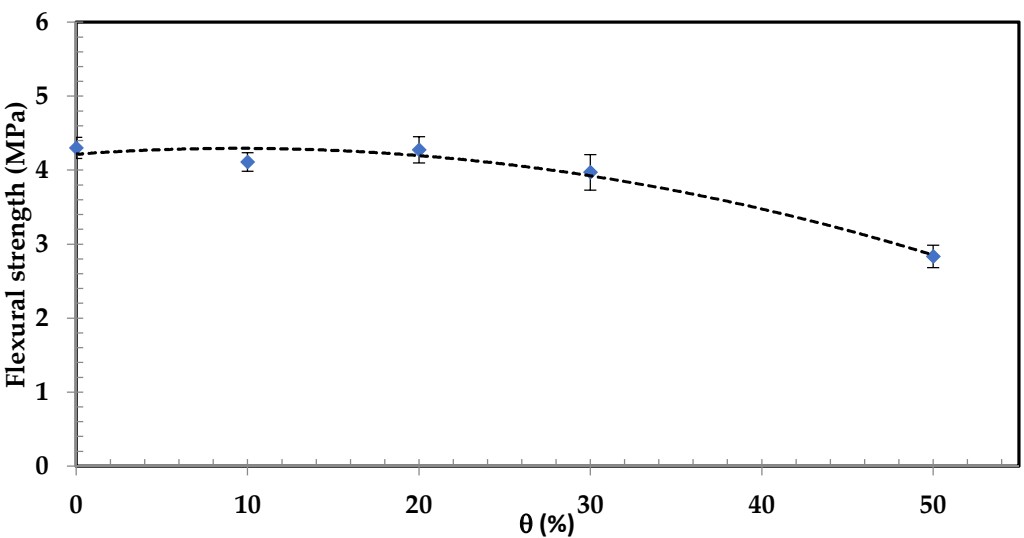

**Figure 13.** Flexural strength influence of concrete reinforced with different volume percentages of RWG.

### 3.4. Experimental Results of Thermo-Physical Properties

### 3.4.1. Density

The density variation of concrete reinforced with different volume percentages of RWG varying from 0 to 50%, at 28 and 90 days of curing samples is illustrated in Figure 14. At these two ages, the dry density decreased respectively by around 5% with respect to RC. This gain in lightness is mainly due to the density of RWG which is lower by about 3% than that of sand. It is of 2500 kg.m$^{-3}$ for RWG and of 2680 kg.m$^{-3}$ for sand. The same behavior was observed by [12,15,19] who substituted sand with glass with the same size in concrete.

### 3.4.2. Thermal Conductivity

Thermal conductivity characterizes the ability of a material to conduct heat. The more thermal conductivity is low the more thermal performance of a material is significant.

Figure 15 displays the thermal conductivity of composite at dry and saturated states as function of RWG volume content with an average measurement error for all samples respectively of 5% and 7%. One can obviously observe that the thermal conductivity decreased with the increase of the RWG volume content in concrete at both dry and saturated states. At dry state, the thermal conductivity dropped from 1.48 W.m$^{-1}$K$^{-1}$ for RC to 0.95 W.m$^{-1}$K$^{-1}$ for RWG50, representing a reduction of about 36%. This result is due to the thermal conductivity of recycled waste glass which is 38% lower compared to the one of natural sand 0.463 W.m$^{-1}$K$^{-1}$ [56]. The same behavior was reported in [57] as the thermal conductivity of mortar-based composite decreased with the natural fibers

weight content increase; this is due to the fact that the natural fibers are known to have lower thermal conductivities than the ones of mortar solid matrixes.

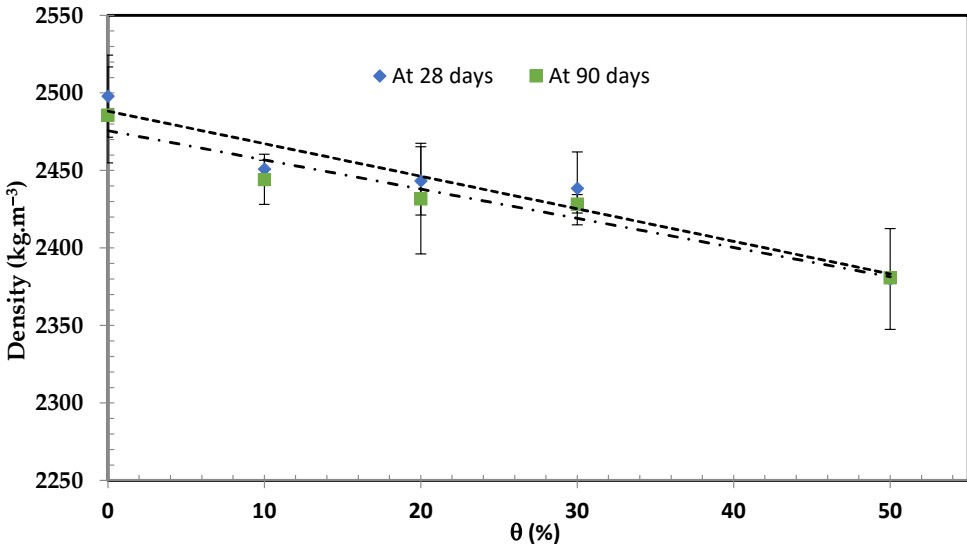

**Figure 14.** Density variation of concrete reinforced with different volume percentages of RWG at 28th and 90th days.

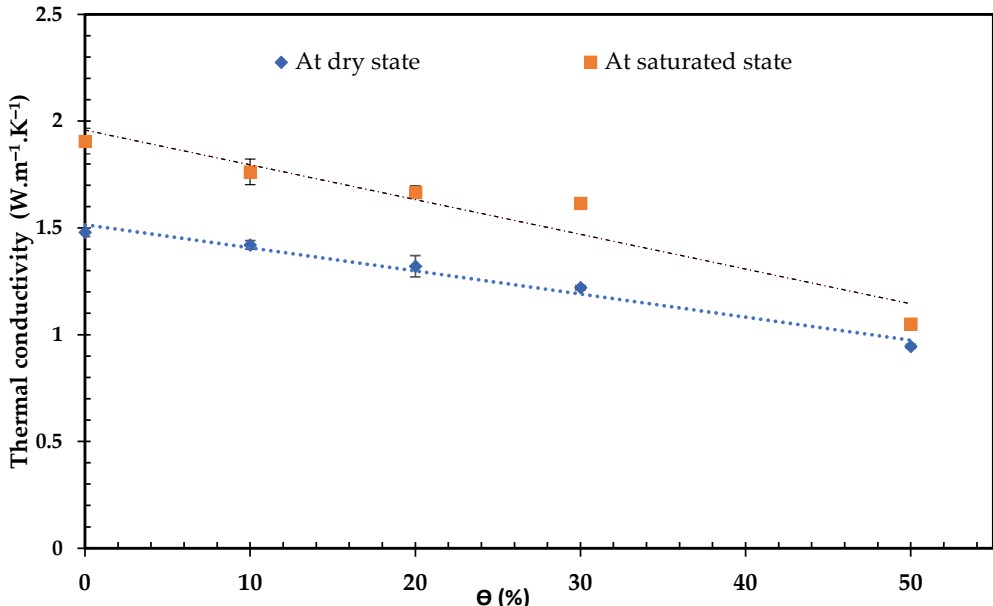

**Figure 15.** Thermal conductivity variation of concrete incorporated with different volume rates of RWG, at dry and saturated states.

At saturated state, the thermal conductivity also decreased by around 44% as RWG is further incorporated into concrete. For each volume percentage, the thermal conductivity value at saturated state is greater than the one at dry state. This difference diminishes slightly from 22.37% to 22.32% respectively for RC and RWC50. These findings can be explained by the fact that the air in pores is replaced by water and thermal conductivity of water (0.601 $W.m^{-1}K^{-1}$) is higher than that of air (0.026 $W.m^{-1}K^{-1}$) [58].

### 3.4.3. Thermal Effusivity

Thermal effusivity characterizes the ability of a material to exchange an important amount of thermal energy in a short time with the external environment in an unsteady

state of heat flow. The results of this quantity at dry and saturated states with average measurement errors for all samples respectively less than 6% and 3% are presented in Figure 16. It can be seen that the thermal effusivity of concrete decreased with the volume content increase of RWG in concrete by about 8.06% and 21.28% respectively at dry and saturated states. These results can be attributed to the fact that air in pores is replaced by water and the thermal effusivity of water (1588 J.K$^{-1}$.s$^{-1/2}$.m$^{-2}$, at 20 °C) is higher than that of air (6 J.K$^{-1}$.s$^{-1/2}$.m$^{-2}$, at 20 °C) [59]. Moreover, the thermal effusivity value at saturated state is greater than the one at dry state, similarly to thermal conductivity. This difference diminished significantly from 19.04% to 5.45% respectively for RC and RWC50. These findings could also be related to the absorptivity of the studied composite which deceased with the increase volume content of RWG in concrete, due to the nonabsorbent characteristic of glass (Table 2). This characteristic leads to the absorption decrease of the composite as the RWG is incorporated in concrete [60], which in turn affects positively the thermal effusivity of the studied sample.

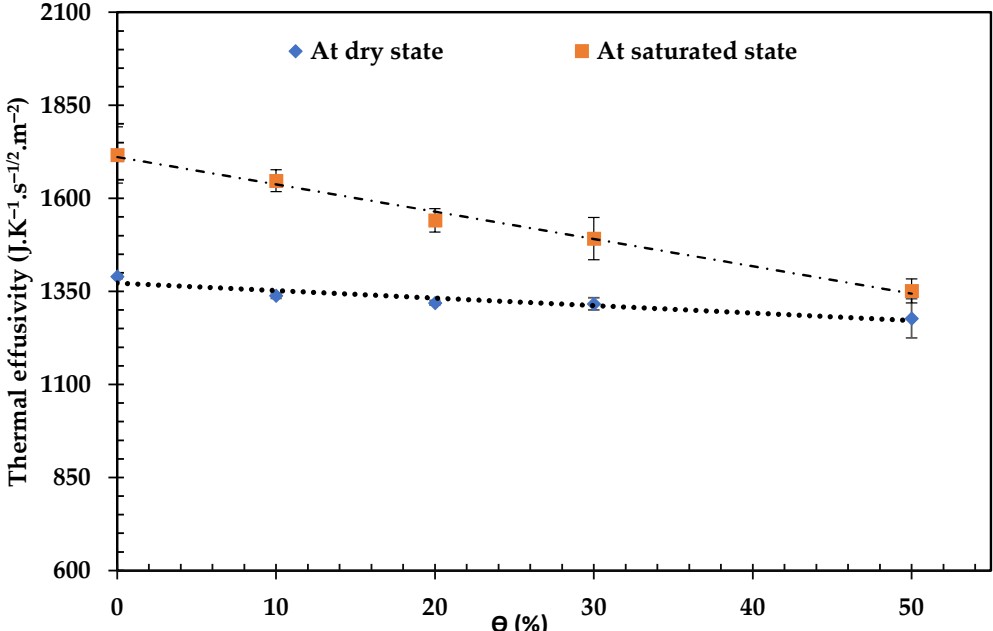

**Figure 16.** Thermal effusivity variation of modified concrete as function of RWG volume rate at dry and saturated states.

### 3.4.4. Thermal Diffusivity and Volumetric Thermal Capacity

The calculated thermal diffusivity and volumetric thermal capacity are depicted in Table 6. Concerning the thermal diffusivity obtained from Equation (6), it decreased with addition of RWG by around 51% for both dry and saturated states as it dropped respectively from $5.17 \times 10^{-7}$ m$^2$.s$^{-1}$ to $1.13 \times 10^{-6}$ m$^2$.s$^{-1}$ and from $1.23 \times 10^{-6}$ m$^2$.s$^{-1}$ to $6.04 \times 10^{-7}$ m$^2$.s$^{-1}$. This decrease is due to the combined effects of thermal conductivity and thermal effusivity as the reduction in thermal conductivity of 36% exceeds the reduction in thermal effusivity of 8% for the highest RWG volume content.

The volumetric thermal capacity indicates the amount of energy required to change the temperature of a given material's volume, and it expresses the amount of heat stored. The results of this parameter calculated from Equation (7) are presented in Table 6; it increased respectively by about 25% and 11% at dry and saturated states. This increase may be due to the increase in volume content of RWG in concrete with a volumetric thermal capacity value of $\rho C_P = 2270$ kJ.m$^{-3}$. K$^{-1}$ that is superior to the one of natural sand $\rho C_P = 1600$ kJ.m$^{-3}$. K$^{-1}$ [61].

**Table 6.** Thermal diffusivity and volumetric thermal capacity of modified concrete.

| Composite | Thermal Diffusivity $\alpha$ (m$^2$.s$^{-1}$) | | Volumetric Thermal Capacity $\rho C_p$ (kJ.m$^{-3}$. K$^{-1}$) | |
|---|---|---|---|---|
| | At Dry State | At Saturated State | At Dry State | At Saturated State |
| RC | $1.13 \times 10^{-6} \pm 3.93 \times 10^{-8}$ | $1.23 \times 10^{-6} \pm 1.86 \times 10^{-8}$ | $1545.75 \pm 185.16$ | $1305.00 \pm 27.57$ |
| CRWG10 | $1.12 \times 10^{-6} \pm 4.39 \times 10^{-8}$ | $1.12 \times 10^{-6} \pm 1.16 \times 10^{-8}$ | $1540.32 \pm 107.51$ | $1261.35 \pm 31.45$ |
| CRWG20 | $1.00 \times 10^{-6} \pm 8.55 \times 10^{-8}$ | $1.17 \times 10^{-6} \pm 0.90 \times 10^{-8}$ | $1424.52 \pm 84.20$ | $1318.66 \pm 62.70$ |
| CRWG30 | $0.85 \times 10^{-6} \pm 3.52 \times 10^{-8}$ | $1.17 \times 10^{-6} \pm 1.03 \times 10^{-8}$ | $1377.51 \pm 113.30$ | $1420.63 \pm 46.58$ |
| CRWG50 | $5.17 \times 10^{-7} \pm 5.65 \times 10^{-8}$ | $6.04 \times 10^{-7} \pm 4.01 \times 10^{-8}$ | $1726.83 \pm 160.12$ | $1739.14 \pm 98.98$ |

## 4. Conclusions

This research paper focused on the valorization of RWG by using it as an alternative to scarce sand, in concrete. Different volume percentages of RWG varying from 0 to 50% were incorporated into concrete. Durability against alkali silica reaction and thermophysical and thermomechanical characterization have been experimentally carried out.

The analysis of the SEM-EDS pictures showed that the incorporation of RWG with maximum size that did not exceed 1.25 mm in concrete did not produce deleterious alkali-silica reaction. The optical microscope characterization of MRGW50 indicated that only glass particles with pre-existing micro-cracks were cracked due to the ASR. Furthermore, the ASR tests on the mortar-based composite reinforced with different volume contents of RWG revealed that the expansion activity remained well below the limit expansion value of 0.15%. Regarding the compressive strength results, it increased respectively at 28 and 90 days of curing by up to 1.63% and 29% for 20% of volume content in concrete; beyond 20%, it diminished receptively by 30% and 3.2% due to the low adherence between cement paste and glass particles. It was also noticed that the compressive strength of RWG50 evaluated better with cure age. As for flexural strength, it decreased by around 34% for the highest volume content of RWG, at 28 days. Thereafter, the addition of RWG into concrete led to a gain in lightness of the manufactured composite by reducing its density by around 5%. Furthermore, the thermal conductivity and thermal effusivity decreased respectively by 36% and 8.06% at dry state and they diminished respectively by 44% and 21.28% at saturated state, related to RC. The calculated thermal diffusivity decreased by about 51% at both dry and saturated states, while the calculated volumetric thermal capacity increased by about 25% and 11% respectively at dry and saturated states.

It may therefore be concluded that the elaborated composite by substitution of sand with RWG could be of use in the building construction as structural material. It is durable against the ASR and it can be of great interest in improving the thermal performance of buildings.

**Author Contributions:** Conceptualization, I.A., S.S. and F.-E.E.; methodology, I.A., S.S. and F.-E.E.; validation, I.A., S.S., F.-E.E. and L.B.; formal analysis, I.A., S.S. and F.-E.E.; investigation, I.A. and S.S.; resources, I.A.; writing—original draft preparation, I.A., S.S. and F.-E.E.; writing—review and editing, I.A. and L.B.; supervision, L.B. All authors have read and agreed to the published version of the manuscript.

**Funding:** This research received no external funding.

**Institutional Review Board Statement:** Not applicable.

**Informed Consent Statement:** Not applicable.

**Data Availability Statement:** Data of this paper will be available from corresponding author, L.B., on reasonable request.

**Acknowledgments:** A big thank you to the National Center for Research and Studies on Water and Energy (Cadi Ayyad University, Morocco) for the technical and scientific support to this work.

**Conflicts of Interest:** The authors declare no conflict of interest.

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
