# Peer review of "Effects of Recycled Fine Glass Aggregates on Alkali Silica Reaction and Thermo-Mechanical Behavior of Modified Concrete"

_applsci, doi:10.3390/app11199045_

Round 1

Reviewer 1 Report

This manuscript can meet the requirements for publication in Applied Sciences

Author Response

Comments and Suggestions for Authors

This manuscript can meet the requirements for publication in Applied Sciences

Response:

Thank you.

Reviewer 2 Report

The authors provided detailed research about the effect of recycled waste glass on the mechanical and thermal behaviors of cementitious material, as well as the negative influence of the alkali silica reaction, via prudential experimental effort. Background of the research was sufficiently discussed in the manuscript. Design, procedure and results of the experiment were clearly presented. Throughout discussion regarding the experiment results were provided. This manuscript was nicely written and could be granted for publication after addressing my questions and suggestions below:

My biggest concern about this manuscript is that it may lack of innovative ideas and constructive conclusions, since most of the experimental results could be linked to/match the observation of previous researches, as the authors discussed in the manuscript. The authors should emphasize more on what is unique about ideas behind this manuscript.

Some questions:

To achieve consistent slump value, the authors changed the w/c ratio and the SP fraction in different sample compositions. As the fraction of RWG increases, the w/c ratio decreases (0.45->0.4). However, lower w/c may result in higher compressive strength. Thus the strength increase as RWG fraction increases may not solely due to the effect of the addition of RWG. How could the authors isolate the strength change from the variation of w/c ratio?

When the authors kept the newly casted samples in molds, as showed in Fig. 3 (d), was the open surface of the samples directly exposed to room ambience? Or it was kept in the same environmental chamber as discussed in line 210-211?

In Fig. 9, the authors mentioned that the crack is caused by the influence of ASR on pre-existed microcracking created by glass bottle crushing. How could the authors identify the cracks in Fig. 9 as ASR cracks rather than the crushing micro-cracks. An image of micro-cracks in control RC sample or some equivalent evidence should be provided in the manuscript.

Minor Issues:

Typos: Curs -> Curing in line 63, 66, 257, 343, 355

Bleu - > blue in line 125

Also please describe all the terms in Eqn 6 and 7.

Reviewer 3 Report

I have no substantive comments,
The work is interesting and correctly presented.
The work may be published in the presented form.

Author Response

Comments and Suggestions for Authors

I have no substantive comments,
The work is interesting and correctly presented.
The work may be published in the presented form.

Response:

Thank you.

Reviewer 4 Report

Dear authors,
Thank you very much for the opportunity to review an article that deals with such an interesting topic.
The title and abstract are slightly confusing and rather discouraging further reading. I recommend being more precise and concise.

The introductory paragraph is written logically and moves on to your hypotheses.
However, I lack information on why you researched the thermal conductivity and thermal effusivity of your concretes? what was the point?

The literature seems to be extensive, but not very current.
In that regard, I recommend expanding the literature with articles on other similar and identical recycling options for aggregates in concrete. E.g.:
10.1016 / j.resconrec.2021.105664
10.1016 / j.conbuildmat.2021.124250
10.3390 / ma13235501
and more....

The description of materials and their properties is adequate. I lack a reference to the standards by which you prepared mixtures and samples.
Even tests do not depend on standards - are they just your processes?
You have a standard for some tests - I noticed that.

The results are in line with expectations, but I am surprised by the significant variance in Figure 11 (30 and 50%) - how do you explain this?
It is also a bit confusing that you have a linear increase and the percentage is not linear (you have skipped 40%).
¨
Similarly, we can look at Figures 12 and 13 - where is 40%?
Why is 40% in Figure 14-16? It is confusing.

The conclusions are not written well. They contain many numbers and percentages, but do not show the clear benefits, output and improvements resulting from the article and research.

The article contains many errors in the members, typed, overall not well prepared linguistically.

Figures 3, 5 and 6 are of strange size, ratio and it does not look scientific at all.

Regards

Round 2

Reviewer 4 Report

Dear authors,
your edits are very strong, well-defended, and I agree with them.
Regards,